# Vertebrate Pollination of Angiosperms in the Mediterranean Area: A Review

**DOI:** 10.3390/plants13060895

**Published:** 2024-03-20

**Authors:** Benito Valdés

**Affiliations:** Department of Plant Biology and Ecology, University of Seville, 41004 Sevilla, Spain; bvaldes@us.es

**Keywords:** Mediterranean area, ornithogamy, saurogamy, *Rhamnus*, *Brassica*, *Anagyris*, *Scrophularia*, *Cneorum*, *Euphorbia*, *Rhamnus*, *Cinnyris*, *Sylvia*, *Phylloscopus*, *Podarcis*

## Abstract

For a long time, it was considered that entomogamy was the only pollination mechanism in the Mediterranean area. However, data recorded in this review prove that ornithogamy and saurogamy also take place. With the exception of the nectarivorous *Cinnyris osea* (Nectariniidae) which pollinates the mistletoe *Picosepalus acaciae* in Israel, all birds responsible for the pollination of several plant species in this area are primarily insectivorous, sedentary, or migrating passerine birds, particularly *Sylvia atricapilla, S. melanocephala, Phylloscopus collibita* and *Parus caeruleus*. They contribute, together with insects, to the pollination of *Anagyris foetida*, three species of *Scrophularia* with big flowers, *Rhamnus alaternus*, *Brassica oleracea,* and some other plants. The lacertid lizard *Podarcis lilfordi* acts as a pollinating agent on several W Mediterranean islands, where it effectively pollinates *Euphorbia dendroides*, *Cneorum tricocum,* and presumably *Rosmarinus officinalis* and *Chrithmum maritimum*. The flowers of some other plant species are visited by birds or by *Podarcis* species in the Mediterranean area, where they could also contribute to their pollination.

## 1. Introduction

The first plants to colonize the emerged lands (Bryophyta and Pteridophyta) [1] still depend on water for their reproductive processes, but they use the air to disperse the spores. Air is the dispersal agent of the male spores, now called pollen grains, of most Gymnosperms, which originate from heterosporous Pteridophyta [2]. In the Mesozoic period, a group of Gymnosperms evolved to give rise to Angiosperms or flowering plants [3], the most evolved and diversified plant group, with about 300,000 species [4], which are the dominant components of most plant formations that cover the emerged lands. Although some groups of Angiosperms use water or air as a means to transport their pollen grains, over 87% of them take advantage of animals [5] for more efficient transportation of pollen from the male organs where they are formed (stamens of male or hermaphrodite flowers) to the stigma of the carpels of female or hermaphrodite flowers, though a process known as pollination.

Angiosperms had an authentic evolutionary explosion in the Cretaceous period [6], largely coincident with the greatest evolution of insects with complete metamorphosis [7]. Primarily used by Angiosperms as pollination agents, whose mutualistic relationships are responsible for the high diversity of flower structures, colors and smells used to attract pollinators; the rewards offered to visitors are primarily nectar and pollen. Later on, several groups of vertebrates, particularly birds, bats, and lizards, mainly on oceanic islands, will begin to participate in the pollination processes.

Entomogamy, pollination by insects, occurs all around the world. Pollination by birds (ornithogamy) and mammals, particularly by bats (cheiropterogamy), is common in America, Africa, Asia, and Oceania; cheiropterogamy is mainly in tropical and subtropical areas, and pollination by lizards (saurogamy) is relatively common on oceanic islands.

The content of this paper demonstrates that ornithogamy and saurogamy, even if not common, can also play an important role in the pollination of several plants in Europe in general and in the Mediterranean area in particular. No examples are known, however, of pollination by bats in this area.

## 2. Ornithogamy

When we think about pollination by birds, the first thought that comes to mind is hummingbirds, which are highly specialized pollination agents. They make up the Trochilidae family, which is endemic to America. They are almost exclusively nectarivorous. Small in size and weight, they have a long and straight beak quite appropriate to take nectar from the bottom of tubular or narrowly infundibuliform corollas, which are brightly colored, mainly in red tones, and are particularly attractive to birds. While hovering to suck nectar from a flower, their beak and head come into contact with the anthers and stigmas of the flowers; pollen grains adhere to the beak or feathers of the forehead, chin, or throat of the bird, which can transfer them to the stigma of another flower in the next visit, carrying out pollination.

In the old world, three main groups of specialized nectarivorous pollinating birds play the same role as American hummingbirds: sunbirds (family Nectariniidae) who live from Africa to Australia mainly in tropical areas, honey eaters (family Meliphagidae) of Australia and Oceania, and lorikeets (tribe Lorinii of family Psittaculidae) of SE Asia and Oceania [8]. The birds of the first two groups have long downward curved beaks, while the third, made up of small parrots, have short beaks with hooked upper mandibles. While hummingbirds reach nectar hovering, sunbirds, honey eaters, and lorikeets get it by perching, but rarely by hovering.

Diverse passerine birds, not particularly nectarivorous but omnivorous, insectivorous, or even frugivorous, drink nectar as part of their diet by regularly or occasionally visiting flowers. While searching for nectar, they can contact the reproductive organs of the flower. Pollen, often mixed with nectar, can adhere to their beaks or feathers and is transferred to the stigma of another flower, performing pollination. They all have short beaks and perch to reach nectar through the corolla opening or directly when flowers are flat, and nectar is easily accessible. However, in most cases, these birds are nectar thieves, or they are even nectar robbers, according to the terminology of Inouye [9], which pierce flowers to extract nectar instead of entering them.

Although insects are considered to be the main plant pollinators in the Euro-Mediterranean region, in Europe, at least 46 species of passerine birds basically insectivorous visit 95 native, naturalized, or cultivated plant species [10], figures which can be extrapolated to the Mediterranean area. New data were added by da Silva et al. [11] in a study of pollen carried by birds in a secondary forest in Larça, near Coimbra (C Portugal). However, the role of birds as pollinators has so far been confirmed only in a few cases.

The possibility that a bird could act as a pollinator in Europe was suggested in 1985 by Kay [12], according to observations made throughout the previous years in three localities of Glamorganshire (Wales, UK), as a part of a general study of the pollination biology of *Salix caprea* L. and *S. cinerea* L. In the three localities, blue tits (*Parus caeruleus*) were regular visitors to catkins of both willows and responsible for some interchanges between male and female trees, which led the author to conclude that blue tits could pollinate those *Salix* species during their visits.

However, the first case of effective ornithogamy in Europe was made known by Burquez [13] when publishing the results of studies carried out in 1986, 1987, and 1988 on the visits of blur tits (*Parus caeruleus*) to the flowers of plants of *Fritillaria imperialis* L. (Liliaceae, an ornamental herbaceous plant native to W Asia) growing in the Cambridge University Botanic Garden and in some private gardens around the city. When visiting the flowers, blue tits perch on the main stem just below the overhanging campanulate flowers to insert their head into them to probe the nectaries, which are placed at its base. As blue tits are the only floral visitors to contact the anthers or stigmas, legitimate pollination was verified by comparison of the fruit set: 0% when flowers were visited by insects (*Bombus*) and 47% when visitors were blue tits [13].

Ornithogamy was detected for the first time in the Mediterranean area by Calvario et al. [14]. In 1987, they trapped three blackcaps (*Silvia atricapilla*) in the natural reserve “Bosco di Palo” at Ladiscapoli (c. 40 km W of Rome) and two garden warblers (*Sylvia borin*) on the island of Capri (near Anacapri), with the upper part of their beaks containing considerable quantities of pollen mixed with the nectar of *Rhamnus alaternus* L. (Rhamnaceae), a shrub rather common in the Mediterranean area. When publishing their observations, they gave convincing reasons to consider *Rhamnus alaternus* as an effectively pollinated species by those two mainly insectivorous birds.

However, bird pollination of the Mediterranean bushy leguminosae *Anagyris foetida* L. is the first clear experimentally proven case of ornithogamy in the Mediterranean area. Ortega Olivencia and co-workers demonstrated that, besides being pollinated by insects in the Iberian Peninsula, this plant is also effectively pollinated by three passerine birds: the blackcap (*Sylvia atricapilla*), the Sardinian warbler (*S. melanocephala*), and the chiffchaff (*Phylloscopus collibita*) [15,16]. Their studies were developed over three years in two populations of *A. foetida* of SW Iberian Peninsula (Olivenza area, province of Badajoz, near the border with Portugal) and some distant populations in Andalusia (S Spain) and Valencia (E Spain). They were also detected as very rare visitors, house sparrows (*Passer domesticus*), blue tits (*Parus caeruleus*), and stonechats (*Saxicola horchata*) [15]. Further, Haran et al. [17] suggested that in Israel, in addition to the blackcap and the Sardinian warbler, the white-spectacled bulbul (*Pycnonotus xanthopygos*) and the migrating lesser whitethroat (*Curruca curruca*) can be considered as potential pollinators and the Spanish sparrow (*Paser hispaniolensis*) and an occasional pollinator of *A. foetida*. At the same time, the chiffchaff is best regarded as a non-pollinating or poor-pollinating visitor of this species in Israel. The only obligate nectarivorous bird that occurs in the Mediterranean area, the nectariniidae Palestinian sunbird, is primarily a nectar thief of *A. foetida*. Some migrating passerine birds were also trapped with *A. foetida* pollen on their feathers: the common whitethroat (*Sylvia communis*), the orphean warbler (*S. hortensis*), and the Rüppell’s warbler (*Curruca ruppeli*). However, it seems that their role as potential pollinators is negligible [17].

Ortega-Olivencia et al. [18] added new cases of ornithogamy in the Mediterranean area in three species of *Scrophularia* (Scrophulariaceae) with big (10–20 mm) reddish, orange-reddish or pinkish-purple flowers: *S. sambucifolia* L., *S. grandiflora* DC., and *S. trifoliata* L. The first is native to the SW Iberian Peninsula and NW Africa; the second is endemic to the CW Portugal (Beira Litoral) [19]; the third is endemic to Corsica, Sardinia, and Gorgona islands [18]. The three are effectively pollinated by bees and bumblebees but also by passerine birds. Bird pollination was confirmed by the study of two Spanish populations of *S. sambucifolia* (El Gastor and Morón, in Cádiz and Seville provinces, respectively), two populations of *S. grandiflora* (Rabaçal and Pombalinho, Coimbra district, Portugal), and two populations of *S. trifoliata* (Santu Lussurgiu and Bitti, Sardinia). The three species are pollinated by Sardinian warblers (*Sylvia melanocephala*) and blackcaps (*S. atricapilla*) (estimate for *S. trifoliata*), and *S. sambucifolia* also by chiffchaffs (*Phylloscopus collibita*).

Cecere et al. [20] assumed that in C. Mediterranean, migrating birds of several species of *Sylvia* may pollinate the flowers of *Brassica oleracea* L. (Cruciferae), in accordance with observations made in Ventotene, one of the islands in the Tyrrhenian sea (Campania, Italy). Karlioglu Kiliç et al. [21] suggest that the E Mediterranean migrating birds, the lesser whitethroat (*Curruca curruca*) and blackcap (*Sylvia atricapilla*), may contribute to the pollination of several plant species cultivated in the Istanbul area (Turkey), as they consume both pollen and nectar of those plants as part of their diet.

In the Mediterranean area, the only obligate nectarivorous bird that acts as a pollinator is the Palestine sunbird, *Cinnyris osea* (=*Nectarinia osea*) subsp. *osea*, a sedentary member of the family Nectariniidae, whose distribution area is limited to SW Syria, S Lebanon, W Jordan, Egypt (Sinai), W Saudi Arabia, Yemen, and Omar [22]. Vaknin et al. [23] studied flower visitors and fruit sets in a population from N Arava Valley (Israel) of *Plicosepalus acaciae* (Zucc.) Wiens and Polhil (=*Loranthus acaciae* Zucc.), a mistletoe that is distributed throughout NE Africa and the near-east [24]. They found that in summer, the flowers of this mistletoe were visited by a wide spectrum of pollinators, both birds and insects, while in winter, the flowers are visited almost exclusively by *Cinnyris osea*, which results in a higher seed set percentage in winter than in summer. The pollinator of *Plicosepalus acacia,* the Palestine sunbird, is, however, a nectar thief of other plant species, as is the case of *Anagyris foetida,* indicated above.

## 3. Saurogamy

Lizards have been known to be seed dispersers since the beginning of the 20th century [25], but their role as potential pollinators has been revealed more recently. It has been documented only on islands [26].

The first confirmed case of saurogamy in the Mediterranean area is the effective pollination of the endemic circum-Mediterranean *Euphorbia dendroides* L. (Euphorbiaceae) by the Balearic lizard *Podarcis lilfordi*, endemic to the Balearic Islands, where it lives in the archipelago of Cabrera and some islets of the coasts of Mallorca and Menorca [27].

The species of the genus *Podarcis* (Lacertidae) are considered food generalists but primarily insectivorous [28]. However, in the Mediterranean area, climatic conditions with strong fluctuations throughout the year lead to the corresponding variation in food resources and, hence, variations in the diet of many animals. This is why in Mediterranean islands, many *Podarcis* species feed on plants, including their nectar and pollen, at least temporarilly, particularly in winter and spring when insect availability is low [28,29,30].

In the archipelago of Cabrera and in the islet of Moltona (south of Mallorca), Sáez and Traveset [29] observed that in March–April, when sucking the nectar of the native shrub *Euphorbia dendroides*, *Podacis lilfordi* rubs its snout against the anthers of male flowers of the cyathia of the plant, While moving through the inflorescences, the central part of its body brushes against both male and female flowers, promoting pollen transportation between plants, with the possibility of laying the collected pollen from male flowers in the stigma of female flowers, which led these authors to assume that *P. lilfordi* could have a role as pollinator. This was experimentally demonstrated by Traveset and Sáez [30] through the study of pollination by insects and by *P. lilfordi* in one population of *E. dendroides* in the L´Espalmador area (island of Cabrera) undertaken in spring 1995 and 1996. During the flowering peak of this plant (from 21 March to 6 April in this population), cyathia produce abundant nectar. They are visited by insects, mainly flies, wasps (*Ancistrocerus ebusianus*), and bees (*Anthophora balearica*), as well as by the Balearic lizard. They verified that the seed set was significantly high in the plants where visits by the lizards had been more frequent.

In addition, Sáez and Traveset [29] and Traveset and Sáez [30] observed that in the archipelago of Cabrera, *Rossmarinus officinalis* L. (Labiatae) and *Globularia alypum* L. (Globulariaceae) were also visited by *Podarcis lilfordi*, although the frequency of visits was considerably lower than on *Euphorbia dendroides* [30]. When visiting the flowers of *R. officinalis*, a shrub widely distributed in the Mediterranean area and Caucasus and naturalized in Macaronesia [31], lizards introduce the tip of their snout into the corolla tube to reach the nectar accumulated at its bottom. The stigma and the anthers come in contact with the upper part of their head [29]; hence, the possibility that this lizard could promote pollination of this plant is a possibility that has not been experimentally proven so far. However, when visiting *G. alypum* and some other plants also present in the area, the activity of this lizard is merely as a herbivore [29].

Pérez-Mellado and Casas [32] studied a new case of saurogamy in the Balearic Islands. They observed that in the islet of Nitge, close to Minorca, *Podarcis lilfordi* visited plants of *Chrithmum maritimum* L. (Umbelliferae) to suck nectar and eat pollen without destroying the flowers. While doing so, pollen adheres to the belly, throat, and lips of the lizards, which are able to transport pollen between plants. Comparing the seed viability of plants visited by *Podarcis* with those that were not visited, the authors concluded that this lizard could be considered an important pollinator of *Chrithmum* in that islet.

A new clear example of saurogamy in the Mediterranean area derives from the studies by Fuster and Traveset [33] developed in Sa Dragonera, an islet close to the western coast of Mallorca (Balearic Islands). Through detailed floral biological experiments, they demonstrated that *Podarcis lilfordi* actively visits the flowers of the *Cneorum tricoccon* L., a shrub endemic to W Mediterranean [34], searching for nectar during the maximum flowering intensity, which takes place during March and April, and contributes significantly to its pollination.

Furthermore, in a plant community on the island of Cabrera, Romero-Egea et al. [35] have found that *P. lilfordi* interacts with 44 plant species. The relationship of this lizard with 13 of them is merely one of florivory. However, legitimate visits, in which the lizards have contact with the reproductive organs of the flowers without damaging them, have been confirmed for 26 species and result in the transportation of pollen amongst plants, which could produce effective pollination. The most frequently visited species are *Daucus carota* L., *Euphorbia dendroides* L., and *Lobularia maritima* (L.) Desv., followed by *Cistus monspeliensis* L., *Lavatera arborea* L., *Lomelosia cretica* (L.) Greuter and Burdet, and *Paronychia capitata* (L.) Lam. Therefore, the extent of possible plant pollination by *Podarcis lilfordi* opens a wide range of possibilities that should be explored.

## 4. Discussion

The studies and observations mentioned above demonstrate that in the Mediterranean area, the incidence of vertebrate pollination could be more widely generalized than so far known. They allow us to say that, so far, the proven or at least most probably bird-pollinated plants in this area are *Rhamnus alaternus*, *Anagyris foetida*, *Scrophularia sambucifolia*, *S. grandiflora*, *S. trifoliata*, *Brassica oleracea*, and *Plicosepalus acaciae*. None of them depend entirely on birds for their sexual reproduction because they, even *Plicosepalus acaciae* with typically ornithophyllous flowers, are also effectively pollinated by insects.

However, observations on flower visitors, particularly the detection of pollen grains attached to the beak and feathers or found in the feces of different passerine birds, indicate that pollination by birds might be a more generalized process in this area. For instance, the presence or absence of pollen loads in feathers from a total of 14,844 passerine birds migrating in spring from Africa to Europe were analyzed by Cerere and coworkers in 14 stopovers in three Mediterranean countries during a study primarily based on ringing activities [36]. Four of these stopovers were located in NW continental Spain, two in the Balearic Islands (Cabrera and Colom islands), three in coastal areas of continental Italy, four in the Tyrrhenian islands Ponza, Zannone, Ventolene, and Ustica, and one in the Greek island of Antikya. Observations centered on birds already known to feed on nectar during spring migrations. The number of birds analyzed and the extension of the area covered by the study are wide enough to consider that the conclusions could be of general application. The presence of the pollen of several plant species was detected in higher or lower quantities in the feathers of blackcaps, garden warblers, subalpine warblers, whitethroats, Sardinian warblers, icterine warblers, melodious warbles, chiffchaffs, wood warblers, and willow warblers. However, they studied nectar feeding only on the island of Ventotene [37], where visited plants were almost exclusively *Ferula communis* L. and *Brassica oleracea* group. However, some birds also visited the native *Malva sylvestris* L. and *Lavatera arborea* L., as well as the cultivated *Prunus avium* L. and *Pittosporum tobira* (Thumb.) W.T. Aiton. These are all spring-flowering species, a season in which insects are not frequent, which can force these birds, primarily insectivorous, to search for nectar and pollen to restore energy.

Cerere et al. [38] made it known that spring migrant passerine birds staging at Ventotene island carried, stuck on their feathers and beak, although sometimes in small quantities, the pollen of many plants, components of the local flora, or natives of distant areas. Some of them are wind-pollinated, such as, for instance, Gramineae, Urticaceae, Chenopodiaceae, Cupresaceae, *Pinus*, *Quercus*, or *Fraxinus* species, and birds visit them to feed on nectar and/or pollen.

This leads us to pay attention to another consideration: the role that migrant passerine birds could play in pollen transfer between more or less distant populations of the same species and, therefore, promote gene flow in plants that reproduce by out-crossing and even between different plant species, which could promote hybrid formation. However, one of the limiting factors for this gene flow is the time pollen grains can keep their viability after being released by the anthers. In other words, the ability to perform the function of delivering the sperm cell to the embryo sac [39]. Further, when pollen is transferred by migrating birds, the incidence of potential gene flow will greatly depend on the time spent by birds to reach their final destination from the starting migration area or from one stopover to the next, as pollen viability may last hours or days, even over one month, according to the species [40], and also the several factors that affect pollen viability [41], including drying. However, how migratory birds divide their time between flight and stopover is still not well known, but it is clear that some passerine birds can cover over 200 km in only one day [42]. Consequently, pollen flow may be possible, but not easy, through bird pollination, and this is a biological process that should not be neglected.

With the exception of the nectariniidae *Cinnyris osea*, birds responsible for plant pollination in the Mediterranean area are not specialized nectarivorous but mainly insectivorous, which suck nectar as part of their diet, particularly during the seasons when insect availability is low, which forces them to search for other food resources. This is also the case with *Podarcis lilfordi*, the only lacertid lizard that is known to pollinate some plant species in the W Mediterranean. The most frequent passerine birds whose pollination activity is known to be more widely generalized in the Mediterranean area are *Sylvia atricapilla*, *S. malanocephala*, and *Phylloscopus collibita*. Two of them (*S. atricapilla* and *S. melanocephala*) are also part of the group of birds that in the Macaronesian archipelagos regularly pollinate at least 12 endemic or introduced plant species [18,43,44,45,46,47,48,49,50], some of them with clear ornithophylous flower syndromes. This seems to indicate that those, and presumably other passerine birds, could also act as pollinating agents in the Mediterranean countries Morocco, Algeria, and Tunisia, which occupy most of the geographic area that separates the Mediterranean and the Macaronesian regions.

With regard to saurogamy, in the Mediterranean islands, lacertid lizards are represented by 32 species, of which 16 belong to the genus *Podarcis* [51]. As herbivory in lacertid lizards is associated with insularity [52,53] and herbivory has been confirmed for several Mediterranean species of *Podarcis* [28,51,52,53,54,55,56], attention must be paid to the dietary behavior of Mediterranean lizards, which will probably lead to discovering new clear examples of saurogamy.

## 5. Concluding Remarks

Besides entomogamy, pollination by vertebrates, although not frequent, also occurs in the Mediterranean area.

In the Mediterranean area, several omnivorous or insectivorous passerine birds, while sucking nectar, can achieve effective pollination of some plant species.

In the W Mediterranean islands, one lacertid lizard acts as a pollinating agent for some plants, a process that is expected to happen in other islands of this area.

Attention should be paid by field biologists involved in the study of trophic networks in the Mediterranean area to detect new cases of ornithogamy and saurogamy and to study the potential extent of gene flow by means of pollen transport by birds.

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
