# Peer review of "Vertebrate Pollination of Angiosperms in the Mediterranean Area: A Review"

_plants, 2024, doi:10.3390/plants13060895_

Round 1

Reviewer 1 Report

Comments and Suggestions for Authors

The author reviews the literature associated with bird and lizard pollination of flowerd   in the Mediterranean area.   This review does highlight the major studies of either experimental work or observations made of bird and lizards as potential, or confirmed, pollinators.  It is important in that it encourages ecologists and pollination biologists to be more aware of the potential pollination behavior of these vertebrates.

In general,  it is fairly well organized and covers the literature available.  There are a lot of awkward sentences, in particular, awkward English structure, and numerous misused propositions.  Some of these are noted below, but it is probably not exhaustive, and I may have missed a few.   These make it a bit difficult to read.   These could be easily tidied up to make it more readable.   There are a lot of non-italicized genus and species names in the main body of the text.

Specific comments

Introduction.

The first paragraph is not really relevant and could be eliminated.

Par 2-  There should be several references listed to the claims made throughout this paragraph.

Lines 33-34,  There are a couple of groups of Gymnosperms – Gnetales and Cycadales that are insect pollinated, and some with very specialized pollination systems,  so this sentence should be corrected or  to state that most Gymnosperms are wind pollinated with the exception of …

Some small grammar issues- lin 31 (‘from’ should be’ on’ ). Line 38 awkward

Par 4, Awkward 2nd sentence.’ , the latest mainly’-  What does this refer to? Not clear

Line 52  ‘in’ should be ‘on’  - change the preposition.

First sentence of Par. 5, The meaning of  ‘ even if not genearalized’      is not clear.

For the rest of the paper, there are many misused prepositions,  and  I do not point out all of them.

Ornithogamy

Par 1 line 1-  the sentence opens with mutualism between plants and birds, but the rewards to seed dispersal and pollination are not given- so only referring to what the plant gets out of the relationship.   Also, ‘are established’ is not quite the correct terminology here.

Par 2. Some small grammar errors – lines 81, 82,84

Par 3, use common name of Parus caeruleus  before its genus/species mention because it is easier for the reader to remember it.  

Line 110- what were the seed set differences in the bagged and unbagged flowers?   What about insect involvement?

Par 4, line 127, ‘rear’ should be ‘rare’?  Sentence is a bit awkward.

Par 5, line 160-  this part of the sentence is awkward, as is the last sentence of the par.  Perhaps change to ‘ The true pollinator of ….. sunbird, is, however a nectar thief of other plant species, for example Anagryus foetida. ‘   

Saurogamy

Par. 1 Is this sentence true?  Are there examples outside of oceanic islands?   I think there is at least one example in South Africa, Drakensburg.

Some misuse of words, line 180 and 208

Par 4, lines 187 .  I could not find meaning of the word ‘cyatia’   which is mentioned twice.

Line 190,  ‘suppose’ do you mean ‘propose’ ?

Par that starts on line 223-  Do you mean ‘ In addition’  or ‘However’  instead of ‘Besides’

Line 226, ‘confirmed’   Sentence is a bit awkward.

Line 229 ‘Lobularia’

Line 232  ‘convenient’ is not clear- , maybe just end sentence with ‘should be explored’

Discussion

Some misuse of prepositions, line 239, 241, 243, 250

Sentence starting in 267 is awkward.

Line 280 – misspelled  ‘through’

Comments on the Quality of English Language

I have mentioned a few places where english is a awkward.   I think most of them can be remedied fairly easily.   

Author Response

CORRECTIONS. REVIEWER 1

  • The English of the manuscript has already been corrected by a British philologist (J. Boyle), and your corrections, in any case, added.
  1. Paragraph 1 (lines 22 to 28) deleted, as it is not relevant to the paper.
  2. Par. 2. Corrected and appropriate references added, as requested.
  3. Line 33. The sentence has been changed to “Air is the dispersal agent of male spores, now called pollen grains, of most gymnosperms, which originated from heterosporous Pteridophyta [2]”. Indeed, Cycadales and Gnetales are insect pollinated (at least visited by insects).
  4. Line 49, “this is” deleted.
  5. Line 51. I have changed “the latest” to “cheiropterogamy”, to make the sentence clear.
  6. Line 54, “generalized” changed to “common”.
  7. The first sentence (lines 58-59) deleted. Rewards are indicated in par. 3 of the introduction (line 46).
  8. 3. I have added “blue tits” before (Parus caeruleus), as requested.
  9. Line 110. I have changed the sentence “by comparison of fruit production in bagged and unbagged flowers” to “by comparison of fruit set: 0% when flowers were visited by insects (Bombus) and 47% when visitors were blue tits [10: 338].
  10. Line 129. “rear” corrected to “rare”.
  11. Line 173. I have deleted “oceanic”, as all examples I give occurs in Mediterranean islands (Macaronesia is not included in the paper), and indeed, the Mediterranean is not an ocean. All cases I know occurs in islands; I ignore where saurogamy has been detected in Drakensburg.
  12. Lines 187 and 194. Cyathia are the characteristic inflorescences of genus Euphorbia.
  13. Line 190. It was just a supposition, not a confirmation (they though that lilfordi could act as a pollinator). I have changed “suppose” for “assume”.
  14. Line 198. I have changed “Besides” to “In addition”.
  15. Lobularia corrected, and all Latin names from lines 228 to 266 are now italicized.
  16. Line 232. I have deleted “convenient to”.

Reviewer 2 Report

Comments and Suggestions for Authors

I am not an expert on pollination in this geographical region, but I have studied insect pollination in northern Europe. I agreed to review out of interest and indeed I found it quite illuminating. It serves a useful purpose in summarising available information. 

Some minor improvements are required. In places further citations are needed. I have noted these in comments on the attached PDF version.

I question whether the first paragraph is required. It could be deleted and the next paragraph edited to form the introductory section. 

Comments on the Quality of English Language

The English requires some improvement. I have attached a PDF with comments and suggested improvements.

There are some issues of formatting that need to be addressed, such as use of italics for species names and style of et al. in citations through out the manuscript.

Author Response

CORRECTIONS. REVIEWER 2. (His/her corrections were made directly in the manuscript)

  • The English of the manuscript has already been corrected by a British philologist (J. Boyle), and your corrections, in any case, added. All Latin names are now in italic font.
  1. Lines 12 and 14, corrected.
  2. Paragraph 1 (lines 22 to 28). Deleted, as requested, as it is not relevant to the paper.
  3. Paragraph 2, corrected and appropriate references added, as requested by another reviewer.
  4. Lines 48 and 52, corrected.
  5. Line 45, “began to”, added.
  6. I do not know any European hummingbird.
  7. Line 79. References [12-18, 20] added, as requested.
  8. Lines 84-85, partially deleted.
  9. Line 92, changed as indicated.
  10. Line 156, corrected as indicated.
  11. Line 173. I have deleted “oceanic”, as all examples I give occurs in the Mediterranean islands (Macaronesia is not included in the paper) and, indeed, the Mediterranean is not an ocean.
  12. Line 208. I have added “demonstrated”.
  13. Line 225, “one”, added.
  14. Line 238, “enterily”, added.
  15. Line 250, “as feeding nectar” is changed to “to feed on nectar”.
  16. I have divided paragraph from lines 262 to 266 into two sentences, as requested. The second: “Some of them are wind pollinated … or Fraxinus species, and birds visit them to feed on nectar and/or pollen”.